# Amorphous Pterostilbene Delivery Systems Preparation—Innovative Approach to Preparation Optimization

**DOI:** 10.3390/pharmaceutics15041231

**Published:** 2023-04-13

**Authors:** Natalia Rosiak, Ewa Tykarska, Judyta Cielecka-Piontek

**Affiliations:** 1Department of Pharmacognosy, Faculty of Pharmacy, Poznan University of Medical Sciences, 3 Rokietnicka St., 60-806 Poznan, Poland; nrosiak@ump.edu.pl; 2Department of Chemical Technology of Drugs, Poznan University of Medical Sciences, 6 Grunwaldzka St., 60-780 Poznan, Poland; etykarsk@ump.edu.pl

**Keywords:** pterostilbene, amorphous solid dispersion, glass transition, Gordon–Taylor equation, Couchman–Karasz equation, molecular modeling, miscibility, DFT calculation

## Abstract

The aim of our research was to improve the solubility and antioxidant activity of pterostilbene (PTR) by developing a novel amorphous solid dispersion (ASD) with Soluplus^®^ (SOL). DSC analysis and mathematical models were used to select the three appropriate PTR and SOL weight ratios. The amorphization process was carried out by a low-cost and green approach involving dry milling. An XRPD analysis confirmed the full amorphization of systems in 1:2 and 1:5 weight ratios. One glass transition (T_g_) observed in DSC thermograms confirmed the complete miscibility of the systems. The mathematical models indicated strong heteronuclear interactions. SEM micrographs suggest dispersed PTR within the SOL matrix and a lack of PTR crystallinity, and showed that after the amorphization process, PTR-SOL systems had a smaller particle size and larger surface area compared with PTR and SOL. An FT-IR analysis confirmed that hydrogen bonds were responsible for stabilizing the amorphous dispersion. HPLC studies showed no decomposition of PTR after the milling process. PTR’s apparent solubility and antioxidant activity after introduction into ASD increased compared to the pure compound. The amorphization process improved the apparent solubility by ~37-fold and ~28-fold for PTR-SOL, 1:2 and 1:5 *w*/*w*, respectively. The PTR-SOL 1:2 *w*/*w* system was preferred due to it having the best solubility and antioxidant activity (ABTS: IC_50_ of 56.389 ± 0.151 µg·mL^−1^ and CUPRAC: IC_0.5_ of 82.52 ± 0.88 µg·mL^−1^).

## 1. Introduction

Pterostilbene (PTR) is a dimethylated derivative of resveratrol which is predominantly found in red wines, blueberries, and several types of grapes [1].

The health benefits of PTR are associated with its antioxidant activity, anti-inflammatory, antidiabetic, and chemopreventive effects. In addition, PTR has the beneficial effect of promoting cell regeneration [1,2]. PTR may play a hepatoprotective role in acute liver failure brought on by different pharmacological drugs, according to preclinical research [3,4]. In a preclinical study, Shi et al. provided evidence that PTR exerts hepatoprotective effects in hepatic ischemia/reperfusion injury [5]. Moreover, PTR may reduce high-fat-induced atherosclerosis by suppressing a number of proinflammatory cytokines, which was confirmed by using a mice model [6]. According to human clinical trials, PTR is safe for use in doses of up to 250 mg/day [7,8]. Riche et al. performed human trials evaluating the dose-ranging efficacy of PTR on metabolic parameters. Researchers also confirmed that PTR reduces blood pressure in adults at 250 mg/day doses [9].

Orally administered PTR shows low bioavailability due to its poor solubility and rapid metabolization [10]. Therefore, there is a need to devise strategies to improve the oral bioavailability of PTR. In the literature, attempts to improve the solubility of polyphenols using various techniques have been reported (including hot-melt extrusion [11], cryomilling [12], freeze-drying [13], spray drying [14] and solvent evaporation). One of the leading methods is the preparation of amorphous dispersions (ASD) with the use of a polymer matrix [15]. The amorphous state of polyphenols is most often stabilized by hydrogen bonds. Recent work shows the amorphous solid dispersion of such polyphenols as quercetin [11,16,17], resveratrol [18], curcumin [18,19,20], naringenin [21], daidzein [22], and genistein [23]. The amorphous form of pterostilbene was obtained, for example, by Tzeng et al. via nanoprecipitation with Eudragit^®^ e100 and polyvinyl alcohol (PVA) [24]. The results obtained by Zhu et al. for pterostilbene-protein nanocomplexes indicated the conversion of crystalline PTR into an amorphous form [25]. The PTR nanoparticles obtained were characterized by an improvement in solubility and a reduction in the proliferation of HepG2 cells. Other methods have also been used to improve the solubility and activity of PTR (such as cocrystals [26,27,28], an inclusion complex [29,30], nanoemulsions [31], and nanoparticles [32]). To date, no one has prepared amorphous solid dispersions of PTR with a polymer that is able to overcome the problems of its poor water solubility.

## 2. Materials and Methods 

### 2.1. Materials

Pterostilbene was supplied by Angene Chemical (London, UK). Soluplus^®^ (polyvinyl caprolactam-polyvinyl acetate-polyethylene glycol graft copolymer) was supplied by BASF SE (Ludwigshafen, Germany). Acetonitrile (high-performance liquid chromatography (HPLC) grade) and formic acid (85%) were provided by POCH (Gliwice, Poland). High quality pure water was prepared using a Direct-Q 3 UV purification system (Millipore, Molsheim, France, model Exil SA 67120). For antioxidant assays 2,2′-azino-bis-(3-ethylbenzothiazoline-6-sulfonic acid) (ABTS) and neocuproine were purchased from Sigma-Aldrich Co. (St. Louis, MO, USA). Ammonium acetate (NH_4_Ac) and methanol were supplied by Chempur (Piekary Śląskie, Poland). Cupric chloride dihydrate was supplied by POCH (Gliwice, Poland).

### 2.2. Characterization of Neat PTR and SOL

#### 2.2.1. True Density of PTR and SOL

A helium gas pycnometer (Accupyc 1340, Micrometrics Instrument Corporation, Norcross, GA, USA) was used to measure the true density of pterostilbene and Soluplus^®^. The AccuPyc 1340 determined the density by measuring the pressure change of helium within the pycnometer. This pycnometer consists of control and analysis modules. Thus, the operational status of the pycnometer can be continually monitored in a status window that is displayed via a computer screen. The sample chamber used in the pycnometer was 10 cm^3^.

#### 2.2.2. Thermogravimetric Analysis (TG) of PTR and SOL

The TG of PTR and SOL were carried out in a TG 209 F3 Tarsus^®^ micro-thermobalance (Netzsch, Selb, Germany). Next, an 85 µL open Al_2_O_3_ crucible was filled with approximately 10 mg of powdered samples. The temperature range for the TG study was between 25 °C and 250 °C, with a constant heating rate of 10 °C·min^−1^ at nitrogen atmosphere (NA). The NA’s flow rate was set at 250 mL·min^−1^. TG data were gathered, and Proteus 8.0 was used to evaluate them (Netzsch, Selb, Germany). The results were visualized with Origin 2021b software (OriginLab Corporation, Northampton, MA, USA).

#### 2.2.3. Differential Scanning Calorimetry (DSC) of PTR and SOL

The DSC analyses were carried out in a DSC 214 Polyma differential scanning calorimeter (Netzsch, Selb, Germany). The reference sample was a blank sealed aluminum DSC pan with a lid, and powdered samples weighing 9 mg were put in sealed pans with a hole in the lid. One heating mode (at a temperature range 25–130 °C and a scanning rate of 10 °C·min^−1^) was used to observe the melting point of PTR in the neat compound and confirmed the amorphous state of SOL. The glass transition (T_g_) of PTR and SOL was observed in melting and cooling modes. The optimized parameters of the melting and cooling modes allow for the observation of glass transition (T_g_) of PTR (↑25–110 °C, 10 °C·min^−1^; →110 °C, 2 min; ↓110–−50 °C, 40 °C·min^−1^; →−50 °C, 2 min; ↑−50 °C–130 °C, 40 °C·min^−1^), and SOL (↑25–100 °C, 10 °C·min^−1^; →100 °C, 5 min; ↑100–215 °C, 40 °C·min^−1^ →215 °C, 5 min; ↓215–−50 °C, 40 °C·min^−1^; →−50 °C, 2 min; ↑−50 °C–130 °C, 40 °C·min^−1^). The nitrogen atmosphere flow rate was set at 250 mL·min^−1^. DSC data were gathered, and Proteus 8.0 was used to evaluate them (Netzsch, Selb, Germany). The results were visualized with Origin 2021b software (OriginLab Corporation, Northampton, MA, USA).

### 2.3. Optimization of the Selection of PTR-SOL Weight Ratio

PTR and SOL at various weight concentrations (0–70% of PTR) were added to a 2 mL Eppendorf tube and mixed using a vortex mixer for 60 s to obtain the physical mixture. Next, powdered samples of 9–10 mg were placed in DSC-sealed pans with a hole in the lid, followed by the melting and quenching (↑25–215 °C, 40 °C·min^−1^; →215 °C, 2 min; ↓215–−50 °C, 40 °C·min^−1^; →−50 °C, 2 min; ↑−50 °C–130 °C, 40 °C·min^−1^) in the DSC 214 Polyma differential scanning calorimeter (Netzsch, Selb, Germany). The applied stages of heating and cooling allowed for the obtaining of PTR-SOL blends and to observe the glass transition for them.

Theoretical studies was carried out to compare the experimental glass transition temperature (T_g_) of PTR-SOL blends with the theoretical values predicted from the Gordon–Taylor and Couchman–Karasz equations [33,34].

Gordon–Taylor equation:(1)$$T_{gmix}=\frac{w_{1}T_{g1}+Kw_{2}T_{g2}}{w_{1}+Kw_{2}}$$
w_1_, w_2_—weight fraction of pterostilbene and polymer, respectively T_g_.

T_gmix_, T_g1_, T_g2_—predicted glass transition temperature of a binary system; glass transition temperature of pterostilbene, and glass transition temperature of Soluplus^®^, respectively.

K—constant indicates a measure of interaction between two components

K expressed mathematically:(2)$$K=\frac{\rho_{1}T_{g1}}{\rho_{2}T_{g2}}$$

$\rho_{1},\rho_{2}$—the densities of two components (PTR: 1.2597 ± 0.0007 g·cm^−3^, Sol: 1.1799 ± 0.0012 g·cm^−3^). The density of PTR and SOL were measured experimentally with a helium gas pycnometer (Accupyc 1340, Micrometrics Instrument Corporation, Norcross, GA, USA), see Section 2.2.1: True density of PTR and SOL.

T_g1_, T_g2_—glass transition temperature of pterostilbene and Soluplus^®^, respectively.

Couchman–Karasz equation:(3)$$T_{gmix}=\frac{w_{1}T_{g1}+Kw_{2}T_{g2}}{w_{1}+Kw_{2}}$$
w_1_, w_2_—weight fraction of PTR and SOL, respectively

T_gmix_, T_g1_, T_g2_—predicted glass transition temperature of a binary system; glass transition temperature of PTR, and glass transition temperature of SOL, respectively

K—constant indicates a measure of interaction between two components

K expressed mathematically:(4)$$K=\frac{{∆c}_{p2}}{{∆c}_{p1}}$$

∆c_p1_ and ∆c_p2_ is the change in the heat capacity at T_g1_ and T_g2_, respectively.

### 2.4. Preparation of Amorphous Solid Dispersion of PTR

Amorphous solid dispersions of PTR with SOL were obtained by dry milling. To obtain the PTR-SOL physical mixtures (ph. m.) in weight ratios of 1:1, 1:2, and 1:5, the weighting of PTR and SOL (450 mg:450 mg, 300 mg:600 mg, and 150 mg:750 mg, respectively) was undertaken, and they were then added to a 5 mL Eppendorf tube and vortexed for 60 s. A 50 mL stainless steel jar that would suit the MIXER MILL MM 400 (Retsch, Haan, Germany) bearings received 900 mg of each ph. m. and three stainless steel balls (ϕ12 mm). The milling frequency was set to 30 Hz, and the milling time was set to 30 min. The entire grinding time for the PTR-SOL 1:5 weight ratio was 30 min, while for the 1:1 and 1:2 weight ratio it was 30 and 60 min (two cycles of 30 min, followed by a break of 5 min). To prevent the overheating and melting of the sample, a 5-min break was added between the cycles. The milling process was carried out at room temperature. The acquired systems appeared as a homogeneous, fine powder. The powders were stored in a desiccator for further investigation.

### 2.5. Identification of Amorphous Dispersion of PTR

#### 2.5.1. X-ray Powder Diffraction (XRPD)

On a Bruker D2 Phaser diffractometer (Bruker, Germany), diffraction patterns were captured using CuKα radiation (1.54060 Å) at tube voltages of 30 kV and tube currents of 10 mA. With a step size of 0.02° 2Θ and a counting rate of 2 s·step^−1^, the angular range was 5° to 45° 2Θ. The obtained data was analyzed using Origin 2021b software (OriginLab Corporation, Northampton, MA, USA).

#### 2.5.2. Differential Scanning Calorimetry (DSC)

A powdered sample of 9–10 mg was placed in sealed pans with a hole in the lid. One heating mode (temperature range 25–130 °C, scanning rate of 10 °C·min^−1^) was used to observe the melting point of PTR in PTR-SOL physical mixtures. Next, the melting and cooling modes were used to observe the glass transition (T_g_) of the PTR-SOL systems (↑25–90 °C, 40 °C·min^−1^; →90 °C, 2 min; ↓90–−50 °C, 40 °C·min^−1^; →−50 °C, 2 min; ↑−50 °C–130 °C, 40 °C·min^−1^). A nitrogen with a flow rate of 250 mL·min^−1^ was used. DSC data were gathered, and Proteus 8.0 was used to evaluate them (Netzsch, Selb, Germany). The results were visualized with Origin 2021b software (OriginLab Corporation, Northampton, MA, USA).

#### 2.5.3. Scanning Electron Microscopy SEM

SEM images were registered to verify the particle morphology and size of the PTR, SOL, and PTR-SOL systems. A FEI Quanta 250FEG microscope was used under the following conditions: room temperature, low-vacuum mode (70 Pa), and an acceleration voltage of 10 kV. ImageJ software (version 1.53t, Wayne Rasband and contributors, National Institutes of Health, Bethesda, MD, USA) was used to evaluate the particle size.

#### 2.5.4. Fourier-Transform Infrared Spectroscopy (FTIR) supported by Density Functional Theory (DFT) Calculations

The spectra were measured by an IRTracer-100 spectrophotometer with a QATR that holds a diamond ATR module. All FT-IR spectra were measured in the absorbance mode between 400 and 4000 cm^−1^ with a resolution of 4 cm^−1^ and an average of 400 scans per measurement. Using LabSolution IR software (version 1.86 SP2, Shimadzu, Kyoto, Japan), all infrared spectra were collected and then processed (baseline correction, normalize). By contrasting the FTIR peaks of pure samples with those of the produced systems, the results were interpreted.

The theoretical spectra of PTR was obtained by the computational method. The calculations were carried out on a PL-Grid platform (website: www.plgrid.pl, accessed on 15 September 2022) equipped with Gaussian 09 software (Wallingford, CT, USA). The geometry optimization and the vibrational frequency calculations of PTR were performed at the fi. The analyzed molecules’ initial geometry was proposed using GaussView software (Wallingford, CT, USA, Version E01), and the normal modes were inspected visually. The scale factor for the vibrational computations was 0.967.

Origin 2021b software (OriginLab Corporation, Northampton, MA, USA) was used to analyze the data that were obtained.

#### 2.5.5. Molecular Docking

Molecular docking was performed by MGLTools 1.5.6 with AutoDock 4.2 (ADT; Scripps Research Institute, La Jolla, San Diego, CA, USA) [35].

The PTR 3D structure was retrieved from the PubChem database in SDF format (PubChem CID: 5281727; website: https://pubchem.ncbi.nlm.nih.gov/, accessed on 2 December 2022). The SOL monomer structure was constructed in the GaussView (Wallingford, CT, USA, Version E01) program. Open Babel (website: http://openbabel.org, accessed on 2 December 2022) was used to perform file conversions [36]. Two amorphous molecular structures of SOL were built by Xenoview v.3.7.9.0 (available online at www.vemmer.org/xenoview/xenoview.html, accessed on 2 December 2022) [37].

The initial molecular monomer structure of SOL was prepared using GaussView. The open-source molecular modeling software Xenoview was used to generate two amorphous molecular structures of SOL (for two monomers and five monomers). The amorphous cell was built by the Xenoview amorphous builder by varying the rotatable torsions using a rotational isomeric state (RIS) method.

The pterostilbene molecule (in PDBQT format) was loaded using AutoDock Tool’s ligand menu. The torsion tree was defined by choosing the root; the number of rotatable bonds was identified, and the file was saved in PDBQT format. The amorphous structure of SOL (in PDB format) was imported into the workstation, polar hydrogen atoms were added, and the Kollman and Gasteiger charges were applied. The file saved in PDBQT format was next used as a target. PTR and SOL were imported into the workspace in PDBQT format for further simulation processing.

Docking was carried out using AutoDock Vina, by using the PDBQT files of the PTR and SOL and the configuration file containing dimensions of the grid box. The grid spacing was set to default (0.375 Å). The grid box values were centered on the macromolecule. The number of grid points along the x, y, and z dimensions was set as 40 × 40 × 40. The total grid points per map were 64,000. The output was saved in the grid parameter file (GPF) format.

AutoGrid was used to calculate affinity grid values around the target molecule (file in format GLG). Next, the genetic algorithm was set to default (the number of GA runs: 10, population size: 150, the number of energy evaluations equaled 2,500,000, and the number of generations equaled 27,000). The Lamarckian genetic algorithm was used, and the output was saved in docking parameter file (DPF) format. After the generation of the DPF file, the AutoDock was executed. The input data were the Auto-Dock executable and DPF files. The output was a file in DLG format. The generated results (in DLG format) were a log file that provided information about the top ten free binding energy and RMSD values by which pterostilbene binds to the SOL. The results were analyzed from the analyze menu of AutoDock Tools and ranked based on their binding energies. The lowest binding energy of the PTR-SOL system was visualized in AutoDockTools.

### 2.6. Studies of PTR Properties after Introduction into Amorphous Dispersion

#### 2.6.1. PTR Concentration Measurements

High-performance liquid chromatography was used to measure the changes in PTR concentrations using the DAD detector (HPLC-DAD). A Shimadzu Nexera (Shimadzu Corp., Kyoto, Japan) with the following features was employed in this study: a SCL-40 system controller, a DGU-403 degassing unit, an LC-40B XR solvent supply module, a SIL-40C XR auto sampler, a CTO-40C column oven, and an SPD-M40 photo diode array detector. A Dr. Maisch ReproSil-Pur Basic-C18 100 column of 5 µm particle size and 100 × 4.60 mm (Dr. Maisch, Ammerbuch-Entringen, Germany) was utilized for the stationary phase. Acetonitrile/0.1% formic acid (70:30 *v*/*v*) was used as the mobile phase. The mobile phase was vacuum-filtered using a 0.45 µm nylon filter. The experimental parameters were as follows: a 0.5 mL·min^−1^ flow rate, a wavelength of 308 nm, and a column temperature of 35 °C. The injection has a 10 µL and the retention time was 4.275 min.

#### 2.6.2. PTR Apparent Solubility

The apparent solubility of PTR was determined experimentally and determines the change in PTR concentration over time in the volume of the acceptor fluid, which does not limit the amount of PTR dissolution for the concentrations used. For this purpose, excess amounts of PTR and the systems were placed in 5 mL Eppendorf tubes, and 5.0 mL of distilled water was added. All samples were mixed using a vortex mixer for 60 s and were centrifuged in a HERMILE Z 216 MK centrifuge machine (HERMLE Labortechnik GmbH, Germany) (5000 relative centrifugal force (RCF) for 5 min at room temperature). The acquired solutions were filtered using a 0.45 µm nylon membrane filter (Sigma-Aldrich, St. Louis, MO, USA) before being subjected to an analysis for the PTR content using an HPLC method that was created and validated (see Section 2.6.1 PTR concentration measurements). The analysis was performed in triplicate.

#### 2.6.3. Antioxidant Properties of PTR

A 2,2′-Azinobis-(3-Ethylbenzothiazoline-6-Sulfonic Acid Assay (ABTS) and Cupric reducing antioxidant capacity (CUPRAC) assay were used to determine the antioxidant activity of the water solution of samples (PTR, PTR:SOL 1:2 *w*/*w*, PTR:SOL 1:5 *w*/*w*). All procedures were described previously [38]. In short, the assays were performed in 96-well microplates. A change of color of the ABTS cation radicals was monitored spectrophotometrically at 734 nm. The absorbance in the CUPRAC assay was recorded at the wavelength of 450 nm. All tests were done in a Multiskan GO UV reader (Thermo-Scientific, Waltham, MA, USA).

During the ABTS assay procedure, add 0.025 mL of aqueous solutions of PTR, PTR-SOL 1:2 *w/w* and PTR-SOL 1:5 *w/w* with specified concentrations (see Section 2.6.2 Apparent Solubility of PTR) and 0.175 mL of ABTS^•+^ cation radical solution to the wells of the microtitration plate. In parallel, a blank test was prepared in which the test solutions were replaced with distilled water. After 10 min, the absorbance of the samples was measured at λ = 734 nm against water as a reference sample. The antioxidant activity (degree of scavenging of ABTS^•+^ cationic radicals) of the tested samples was calculated from Equation (5):(5)$$\mathrm{the}\;\mathrm{degree}\;\mathrm{of}\;\mathrm{radical}\;\mathrm{scavenging}\;(\%)=\frac{A_{0}-A_{i}}{A_{0}}\cdot 100\%$$
where $A_{0}$ is the absorbance of the control and $A_{i}$ is the absorbance of the sample.

During the CUPRAC assay procedure into the wells of the 96-well plate, 0.050 mL of aqueous solutions of PTR, PTR-SOL 1:2 *w*/*w* and PTR-SOL 1:5 *w*/*w* of the specified concentrations (see Section 2.6.2 PTR apparent solubility) and 0.150 mL of the CUPRAC solution were measured. The plate was incubated with shaking at room temperature, and protected from light for 30 min. Color changes were read at λ = 450 nm.

## 3. Results and Discussion

Polyphenols, which are plant secondary metabolites, are the most common plant-derived bioactive components in our diet. Their valuable biological properties (such as anti-diabetic [39,40], anti-inflammatory [41,42], neuroprotective [42,43], hepatoprotective [44], antifungal [45], antibacterial [46] and antiviral [47]) are limited by poor solubility, which may also lower their bioavailability.

To the best of our knowledge, this work is the first report on preparing amorphous solid dispersions of PTR using dry milling to improve the solubility and antioxidant activity of PTR.

The first stage of the study concerned the selection of the optimal pterostilbene-Soluplus^®^ (PTR-SOL) weight ratio for the obtaining of the amorphous solid dispersion. For this purpose, a thermal analysis supported by mathematical models (Gordon–Taylor and Couchman–Karasz equations) was applied.

During TG analysis, it was confirmed that PTR is stable to ~200 °C, and SOL is stable to ~275 °C (Appendix A). The obtained data allowed the appropriate selection of the temperature range during the DSC analysis.

The miscibility of PTR with SOL, the strength of the interaction between stilbene and the polymer, and the predicted values of the glass transition were all determined using differential scanning calorimetry and mathematical models: Gordon–Taylor and Couchman–Karasz equations. The thermograms of the PTR-SOL blends at various weight concentrations are presented in Figure 1a.

All measured systems are characterized by a single-step thermal event corresponding to the glass transition (T_g_). This fact confirms the good miscibility and compatibility between PTR and SOL. SOL has a higher T_g_ value than PTR, and therefore acts as an anti-plasticizer and increases the overall T_g_ value of the obtained blends.

The experimental T_g_ values were compared with the theoretical values calculated by the Gordon-Taylor and the Couchman-Karasz equations (equations were detailed in the Section 2.3 Optimization of the selection of PTR-SOL weight ratio). Figure 1b displays the experimental and predicted values of T_g_ as a function of the weight % of the pure components. The experimental T_g_ values are greater than the predicted T_g_ values by the Gordon–Taylor equation (positive deviation), and suggested stronger polyphenol-polymer interactions than polyphenol-polyphenol or polymer-polymer interactions. Positive deviation, according to the literature, could mean a strong hetero-nuclear interaction (e.g., H-bonding etc.) [48]. This shows that the systems have the ability to create strong bonds, which increases the T_g_ value.

On the other hand, for the Couchman–Karasz equation, only experimental T_g_ for PTR 60%/SOL 40% blends suggested weak or no interactions [49,50]. The literature suggests that the hetero-nuclear interactions in the system were equal to the sum of the homo-nuclear interactions (PTR-PTR and SOL-SOL) of the pure components [48,49,50].

In recent years, the DSC analysis was increasingly used to predict T_g_ values, miscibility, and the interactions of blends/systems with polymers. For example, Zhu et al. [51], during the DSC analysis, observed one glass transition temperature in the catechin-biodegradable poly(3-hydroxybutyrate)s blends, which confirmed that the blends were amorphous and miscible. In another study, thermal analysis and theoretical T_g_ value prediction (using the Kwei equation) were employed by Yen et al. [52] to demonstrate the existence of advantageous interactions between tannic acid and polyesters, as well as their miscibility. Lee et al. [53] used the same methodology as Yen et al. for the miscibility analysis and the interaction identification of the biodegradable polymer with tannic acid. Next, Lee et al. [54] obtained novel miscible ternary blends of biodegradable polymers, including natural polyphenol. They confirmed the miscibility based on the single T_g_ value observed. Whereas, in our earlier research, we also found that the glass transition values and interaction forces in binary hesperidin systems could be predicted by using the Gordon-Taylor equation [34].

According to the above screening result, three weight ratios (1:1, 1:2 and 1:5) were used to prepare PTR-SOL systems by dry milling. The milling process was carried out according to the procedure presented in Section 2.4: “Preparation of amorphous solid dispersion of PTR”. In these dispersions, the content of PTR varied from ~17% to 50%. The effect of the milling process on PTR amorphization was monitored by XRPD at different milling times (0.5 h and 1 h). The XRPD results (unpublished data) suggest that the polymer ratio and milling time affected the preparation of amorphous PTR. For the system at a weight ratio of 1:1, no amorphous PTR was obtained after 30 min and a 1-h milling process. In contrast, the time of 1 h was sufficient for the 1:2 system to be fully amorphous. In the case of the 1:5 system, amorphousness was obtained after just 30 min.

Two PTR-SOL systems (1:2 *w*/*w* and 1:5 *w*/*w*) were found to be in amorphous states upon XRPD and DSC analyses.

The XRPD analysis was performed to investigate the changes to the degree of crystallinity of the PTR both in physical mixtures and the systems prepared via milling. The X-ray diffractograms of PTR, SOL, PTR-SOL 1:2 and 1:5 *w*/*w* physical mixtures and systems are presented in Figure 2.

The crystalline PTR is characterized by a crystal pattern consisting of a series of well-defined sharp peaks at 10.59°, 11.75°, 14.40°, 15.22°, 15.70°, 17.12°, 17.51°, 17.71°, 18.38°, 19.39°, 20.36°, 21.20°, 23.28°, 25.45°, and 29.39° 2Θ (Figure 2, black line). The diffractogram of SOL (Figure 2, red line) displayed no peaks due to its amorphous nature [34]. The two physical mixtures showed similar diffraction patterns to that of PTR, but with the reduced intensity of the diffraction peaks. The appearance of the characteristic “halo effect” (the complete disappearance of Bragg peaks, which are characteristic of the crystalline form) on the diffractograms of PTR-SOL 1:2 and 1:5 *w*/*w* systems indicate the formation of amorphous PTR. The completely amorphous “halo” of the X-ray patterns in compound-Soluplus^®^ systems also were observed by Rosiak et al. [34], Wdowiak et al. [55], Nandi et al. [56], and Lim et al. [57].

The thermal properties of PTR, SOL, PTR-SOL (1:2 and 1:5 *w*/*w*) physical mixtures, and PTR-SOL 1:2 and 1:5 (*w*/*w*) systems were measured by a differential scanning calorimeter (DSC). DSC thermograms of PTR, SOL, and PTR-SOL physical mixtures recorded in one heating mode are summarized in Figure 3.

PTR exhibited a sharp endotherm around the melting point at 100 °C (confirming the crystalline nature of PTR), the physical mixtures (ph. m.) of PTR and polymer in 1:2 and 1:5 weight ratio exhibited endotherms at 96.5 and 97.7 °C, and confirmed the crystalline nature of PTR in these mixtures. A broad peak around 70 °C observed in the thermogram of SOL (amorphous polymer) corresponds to the transition from a glassy to a rubbery state [58]. The obtained results are consistent with the literature reports which indicate that PTR has a melting point at about 95.4 °C [59]; SOL exists in amorphous form [60], and the compound-polymer physical mixtures display a thermal effect corresponding to the crystalline form of the compound [61].

Contrary to physical mixtures, the PTR-SOL 1:2 *w*/*w* and PTR-SOL 1:5 *w*/*w* systems did not show any endothermic peaks (see Figure 4). In addition, a single glass transition temperature observed in PTR-SOL 1:2 and 1:5 *w*/*w* systems indicate complete miscibility. The obtained results confirm the amorphous state of pterostilbene in these systems. Numerous literature reports confirm that an amorphous phase can lead to the disappearance of the melting effect and the emergence of a glass transition (T_g_). In addition, the literature confirms that a single glass transition observed in the thermogram of an amorphous system indicates complete miscibility [51,52,54,62]. For example, the homogeneity of the mixtures was confirmed on the basis of DSC analysis (one glass transition recorded) for tannic acid+poly(ε-caprolactone) [52], poly(ethylene azelate)+poly(ε-caprolactone)+catechin [54], nimesulide+kollidon VA64 [63].

The experimental T_g_ values of the obtained systems were compared with the theoretical values calculated by the Gordon-Taylor equation and the Couchman-Karasz equation; see Table 1 (equations were detailed in the Section 2.3 Optimization of the selection of PTR-SOL weight ratio).

The free volume theory is the foundation for the Gordon-Taylor and Couchman-Karasz equations. These mathematical models are applicable under the presumption that the mixing partners are perfectly mixed and have equal shapes and sizes. It indicates no interactions between the mixing partners, and that the substances behave additively in their free volume [33]. The differences observed between theoretical and experimental values are related to interactions between PTR and SOL. Higher T_g,exp_ values than T_g,G-T_, and T_g,C-K_ values confirmed positive deviations in PTR-SOL systems.

SEM images were registered to verify the particle morphology and size of PTR, SOL, and PTR-SOL systems (Figure 5a–d).

SEM images of neat PTR (Figure 5a) revealed crystalline, irregular shaped blocks. The particles were not uniform in size (approximately 40–164 μm width and 21–156 μm height). SOL (Figure 5b) showed large, rough, and irregularly shaped particles (a particle size of approximately 212–391 μm width and 274–424 μm height). Micrographs of the physical mixtures revealed crystalline particles of PTR (Figure 5c,e). The particles of the PTR-SOL systems were found to have no sharp edges (Figure 5d,f). The disappearance of free irregular-shaped blocks in PTR-SOL systems suggests dispersed PTR within the SOL matrix and a lack of PTR crystallinity. In addition, SEM micrographs showed that after the amorphization process, PTR-SOL systems had a smaller particle size and larger surface area compared with PTR and SOL. The particles of PTR-SOL systems were not uniform in size, and were approximately 37–266 μm in width, 64–248 μm in height, and 21–98 μm width and 14–95 μm height for PTR-SOL 1:2 and 1:5 *w*/*w*, respectively.

Similar results were presented by Giri et al. for the telmisartan-Soluplus^®^ amorphous solid dispersion [64]. They were confirmed after amorphization changes in surface topography and the disappearance of free needle-shaped structures of telmisartan. Garbiec et al. [65] confirmed via SEM analysis that the amorphous systems of sinapic acid had a reduced particle size after the milling process.

The FT-IR-ATR analysis was employed to identify the interactions between PTR and SOL. The results of the FT-IR-ATR experiment were further validated using molecular modeling.

Appendix A shows the characteristic FT-IR band assignments for the PTR spectra. In order to identify the band’s positions in the PTR FT-IR spectrum, a comparison was made with the theoretical FT-IR spectrum, which was calculated using the B3LYP/6-31G(d,p) level of theory (see Appendix A).

The chemical structure of PTR contains two phenolic rings (described in this paper as rings A and B), one conjugated double bond (described in this paper as the main chain), a hydroxy group at position 4′, as well as two methoxy substituents at positions 3 and 5 (Figure 6a). The structure of SOL contains poly(ethylene glycol), poly(vinyl acetate), and poly(N-vinyl caprolactam) blocks (Figure 6b) [66].

The most intense bands of the crystalline PTR were observed in the FT-IR-ATR spectra at about 800–1600 cm^−1^ and 2850–3370 cm^−1^. In the first range, predominant bands correspond to the –CH wagging, torsion or bending vibration (see Appendix A). The C–C–C stretching/asymmetric stretching was at 1049 cm^−1^, 1061 cm^−1^, 1319 cm^−1^, 1514 cm^−1^, and 1584 cm^−1^, whereas the –OC stretching was detectable at 1049 cm^−1^, 1061 cm^−1^, and 1103 cm^−1^, while the –OH bending was at 1319 cm^–1^. In the range of 2850–3005 cm^−1,^ predominant bands correspond to the –CH_2_ or –CH_3_ stretching in the 3– or 5–OCH_3_ group. The CH stretching at phenolic ring B and the main PTR chain were detectable at 3028 cm^−1^. An intense peak at about 3341 cm^−1^ corresponds to the 4′-OH stretching.

The infrared spectra of the PTR-SOL systems (ratio 1:2 and 1:5) were analyzed and compared with the spectra of the pure component and their physical mixtures (ph. m.) (see Figure 7a–c).

Physical mixtures display bands of both ingredients, suggesting that an interaction between two compounds does not occur. The spectra obtained for the PTR-SOL systems show significant changes in the intensity, shape, and disappearance of many PTR and SOL characteristic bands (see Figure 7a–c). The observed changes are summarized in Appendix A. For example, many characteristic peaks of PTR completely disappeared in the systems (e.g., peaks in the range 800–1100 cm^−1^, 1223 cm^−1^ and 1319 cm^−1^). The changes also affect characteristic SOL bands (shift peak at about 1475 cm^−1^, decreased intensity and/or disappearance of peaks observed in neat SOL at about 1020–1150 cm^−1^, 1634 cm^−1^, 1734 cm^−1^, and 2930 cm^−1^).

FT–IR studies provided evidence that the obtaining of miscible amorphous PTR-SOL systems was possible due to the formation of hydrogen bonds between the C–O, –O–H and/or –CH groups of PTR, and the O–H and C–O ether linkage group of SOL. This is in line with the previous studies that say that, usually, the vibration peak disappearance and/or peak position shifts due to crystal transformation or forming intermolecular hydrogen bonding [67,68]. For example, the formation of a hydrogen bond with the C=O group in SOL was confirmed by Lu et al. for amorphous dispersions of felodipine with Soluplus^®^ [69].

The literature indicates that amorphous systems’ stabilization is often achieved through molecular interactions such as hydrogen bonds. Strong interactions between the polyphenol and polymer resulted in better miscibility and were conducive to obtaining a single-phase amorphous solid dispersion [12]. For instance, Wegiel et al. claims that the best polymers for amorphous solid dispersion of compounds susceptible to intramolecular hydrogen bonding (curcumin was the focus of their investigation) are those that permit the creation of interactions other than hydrogen bonds (e.g., ionic interactions). Kanaze et al. [70] indicated based the presence of hydrogen bonds between the PVP carbonyl groups and hydroxyl groups of naringenin and hesperetin aglycones based on an FT-IR analysis. In another study, Zhu et al. [51] confirmed strong hydrogen bonds between poly(3-hydroxybutyrate)s and catechin in amorphous miscible blends. Next, Lee et al. [54] identified hydrogen bonds between biodegradable polymers and natural polyphenol.

To further verify the above FT-IR-ATR experimental results, molecular modeling was investigated by AutoDock 4.0 software. The possible rotations of PTR’s bonds were determined and shown in Appendix A. The current total of five rotatable bonds of pterostilbene were the bonds between atoms O1 and C7, O2 and C8, C4 and C9, C12 and C10, and C17 and O3.

The visualization of PTR-SOL interactions, as presented in Figure 8, showed that hydrogen bonds stabilize all systems. The results also showed that 3–C–O–C and 4′–O–H groups of PTR and O–H and C–O ether linkage groups of SOL are involved in developing hydrogen bonds in these systems. The suggested simple molecular model was able to appropriately demonstrate both the existence of molecular interactions and the type of experimentally verified interactions. The literature confirmed that molecular modeling helps in the comprehensive understanding of interactions taking place within ASD systems [71,72].

The amorphous solid state offers improved solubility due to the lower energy barrier required to dissolve the molecules. The impact of pterostilbene amorphization on its solubility was tested. The HPLC technique determined concentrations of PTR in the apparent solubility study. The validation parameters are presented in Appendix A. The HPLC analysis confirmed that dry milling to prepare the amorphous form of PTR did not cause compound degradation. In other words, PTR was chemically stable.

The apparent solubility of the pure PTR was far below 4 µg·mL^−1^, classifying these compounds as practically insoluble [73]. Amorphous solid dispersions of PTR led to an increase in apparent solubility. Amorphization improved the apparent solubility by ~37-fold and ~28-fold for PTR-SOL 1:2 *w*/*w* (147 ± 1 µg·mL^−1^) and 1:5 *w*/*w* (111 ± 2 µg·mL^−1^) systems, respectively. The results were strongly related to the amorphous state of PTR, the smaller particle size, the increased surface area, and SOL’s possibility of forming micelles. Rosiak et al. [34] confirmed that amorphous hesperidin-Soluplus amorphous solid dispersions improved the compound’s solubility in water by 301 times. In another study, Garbiec et al. [65] confirmed that the ASD showing the best solubility improvement had the smallest particle size through SEM analysis. Lee et al. [74] confirmed that the amorphous state of chrysin and micelle-solubilizing contributed to the improved solubility of chrysin in solid dispersion formulations. In our earlier work [34], we also suggested the possibility of micelle formation in amorphous hesperidin-Soluplus^®^ systems. In addition, other literature reports have confirmed that amorphous dispersions of polyphenols show increased solubility. For example, Febriyenti et al. [75] attributed the increase in quercetin’s solubility to its amorphous state in a solid dispersion system. Next, Nadal et al. [76] confirmed that amorphous solid dispersions of ferulic acid improved its relative solubility compared to the crystalline form.

It was inferred that the improved apparent solubility might positively affect the antioxidant properties of the compound [34,55,77]. The improvement of PTR solubility after its introduction into amorphous dispersions translated into a change in its biological properties, such as the antioxidant potential. The ABTS and CUPRAC assays were used to determine how the amorphization of PTR affected antioxidant activity. The crystalline form of PTR showed no activity due to practical insolubility; therefore, determining antioxidant properties was impossible. In contrast, the PTR amorphous dispersion significantly increased the antioxidant properties (see Appendix A). The amorphous dispersion of PTR-SOL 1:2 expressed stronger antioxidant activity in the ABTS and CUPRAC assay, with an IC_50_ of 56.389 ± 0.151 µg·mL^−1^ and IC_0.5_ of 82.52 ± 0.88 µg·mL^−1^, compared to PTR-SOL 1:5 (IC50 = 101.842 ± 0.001 µg·mL^−1^ and IC0.5 = 121.00 ± 4.34 µg·mL^−1^). The better antioxidant properties of the 1:2 system compared to the 1:5 system were related to the better solubility of this system.

## 4. Conclusions

As a result of the work carried out, important milestones were achieved: (i) the amorphous solid dispersion of pterostilbene was prepared (with Soluplus^®^ in a 1:2 and 1:5 *w*/*w* ratio), (ii) the possibility of the theoretical prediction of glass transition values and interaction forces in the pterostilbene systems was demonstrated as a result of using the Gordon–Taylor and Couchman–Karasz equations, (iii) we defined the hydrogen bonds between the C–O, –O–H and/or –CH groups of PTR, and the O–H and C–O ether linkage groups of SOL stabilizing the obtained pterostilbene amorphous solid dispersion systems, and (iv) as an effect of the presence of pterostilbene in the amorphous dispersion, an increase in its apparent solubility and antioxidant activity was demonstrated.

The milestones achieved above allow us to suggest that the obtained amorphous solid dispersion of pterostilbene systems are valuable delivery systems that allow for the more effective use of the valuable biological properties of pterostilbene.

## Figures and Tables

**Figure 1 pharmaceutics-15-01231-f001:**
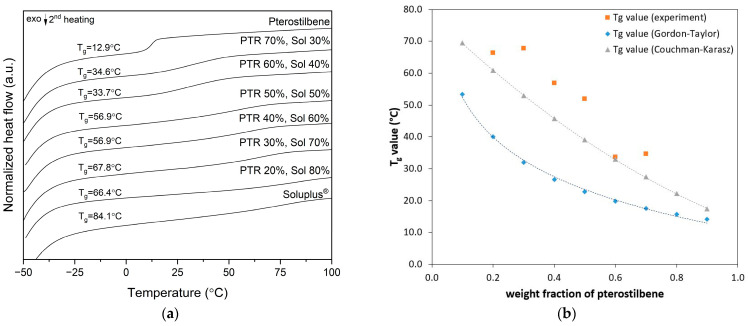
(**a**) DSC thermograms recorded during the second heating scan for pterostilbene-Soluplus^®^ blends; (**b**) Experimental glass transition temperatures for PTR-SOL blends (orange), theoretical glass transition temperatures predicted by Gordon–Taylor (blue), and Couchman–Karasz (grey) equations.

**Figure 2 pharmaceutics-15-01231-f002:**
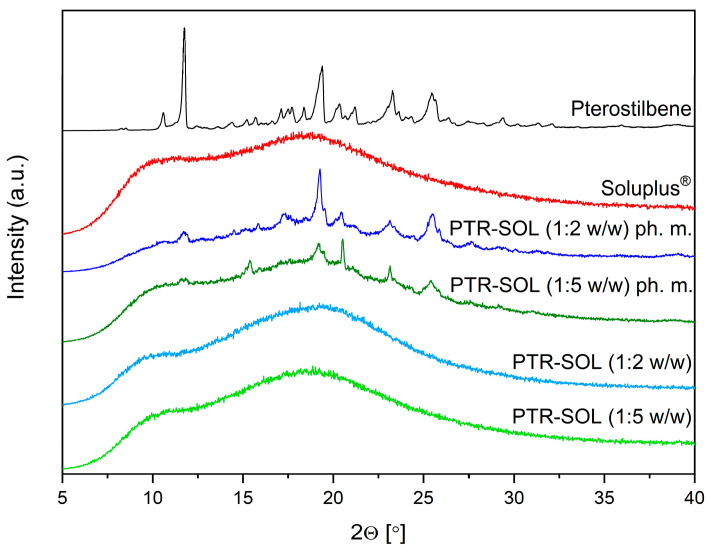
XRPD analysis: pterostilbene crystalline form (black), Soluplus^®^ (red), PTR-SOL 1:2 *w*/*w* ph. m. (dark blue), PTR-SOL 1:5 *w*/*w* ph. m. (dark green), PTR-SOL 1:2 *w*/*w* system (blue), PTR-SOL 1:5 *w*/*w* system (green).

**Figure 3 pharmaceutics-15-01231-f003:**
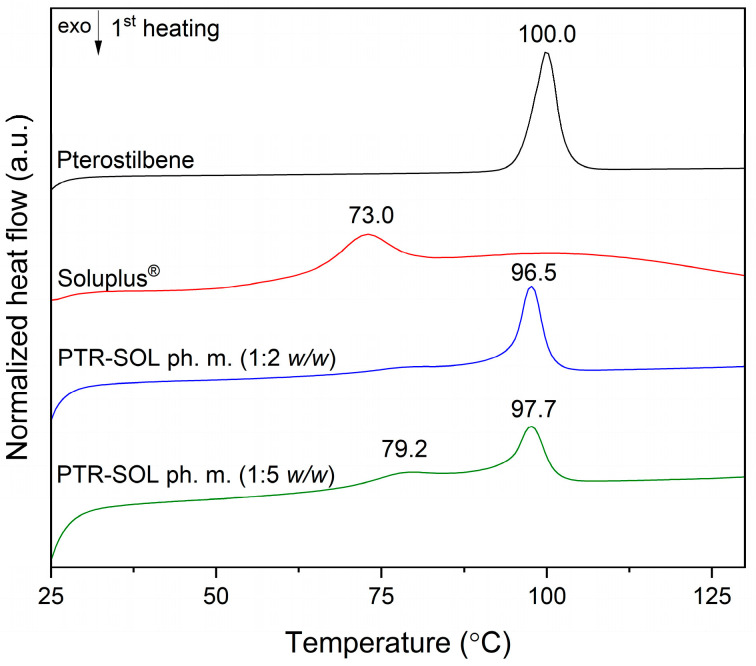
DSC analysis (first heating) of pterostilbene (PTR), Soluplus^®^ (SOL), pterostilbene-Soluplus^®^ 1:2 weight ratio physical mixtures (PTR-SOL 1:2 *w*/*w* ph. m.), pterostilbene-Soluplus^®^ 1:5 weight ratio physical mixtures, (PTR-SOL 1:5 *w*/*w* ph. m.).

**Figure 4 pharmaceutics-15-01231-f004:**
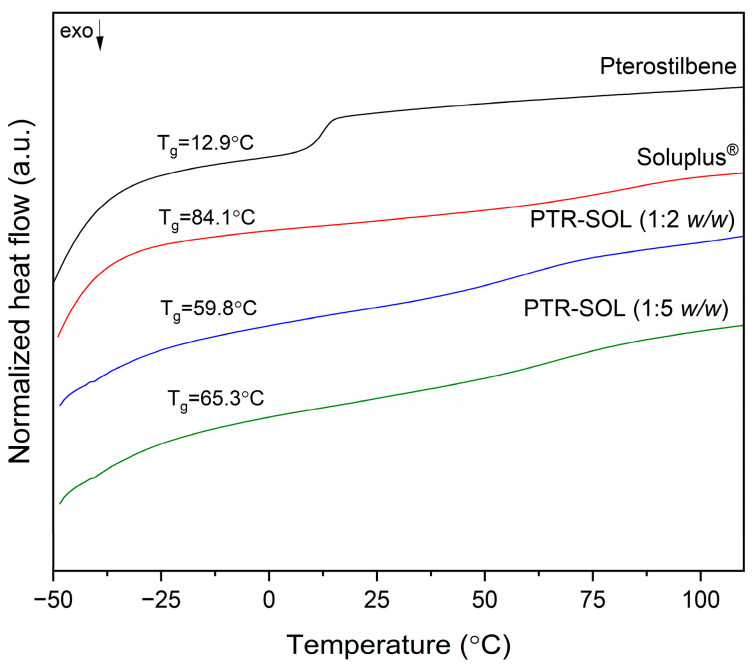
DSC analysis of amorphous pterostilbene (second heating), Soluplus^®^ (second heating), pterostilbene-Soluplus^®^ 1:2 *w*/*w* system (PTR-SOL 1:2 *w*/*w*) (first heating), pterostilbene-Soluplus^®^ 1:5 *w*/*w* system (PTR-SOL 1:5 *w*/*w*) (first heating).

**Figure 5 pharmaceutics-15-01231-f005:**
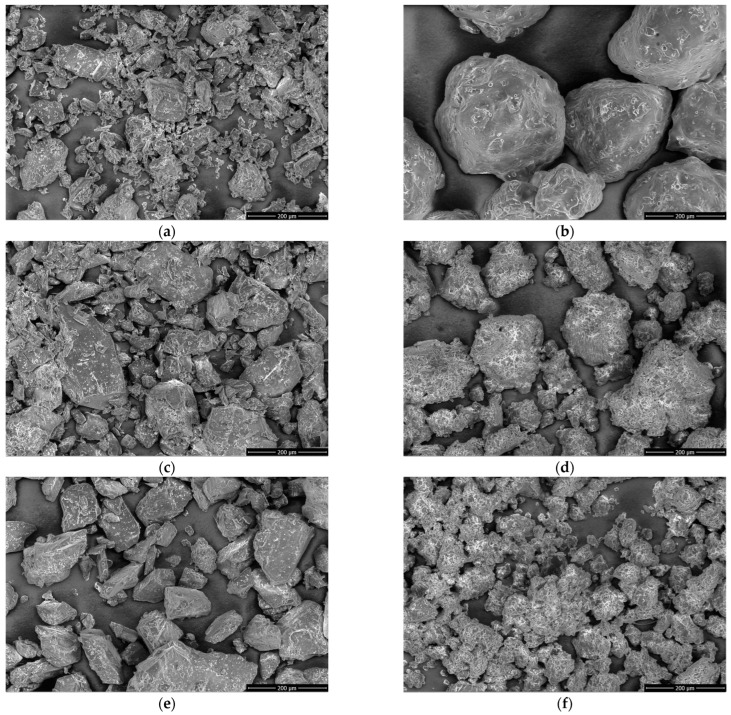
SEM images of pterostilbene (**a**), Soluplus^®^ (**b**), pterostilbene-Soluplus^®^ 1:2 *w*/*w* physical mixture (**c**) pterostilbene-Soluplus^®^ 1:2 *w*/*w* system (**d**), pterostilbene-Soluplus^®^ 1:5 *w*/*w* physical mixture (**e**) pterostilbene-Soluplus^®^ 1:5 *w*/*w* system (**f**).

**Figure 6 pharmaceutics-15-01231-f006:**
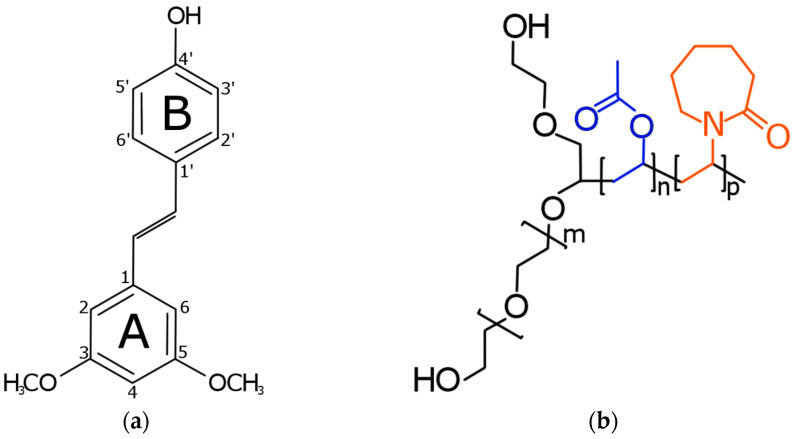
Structure of (**a**) pterostilbene and (**b**) Soluplus^®^ (SOL). The structure of SOL contains poly(ethylene glycol) (black), poly(vinyl acetate) (blue), and poly(N-vinyl caprolactam) (red) blocks.

**Figure 7 pharmaceutics-15-01231-f007:**
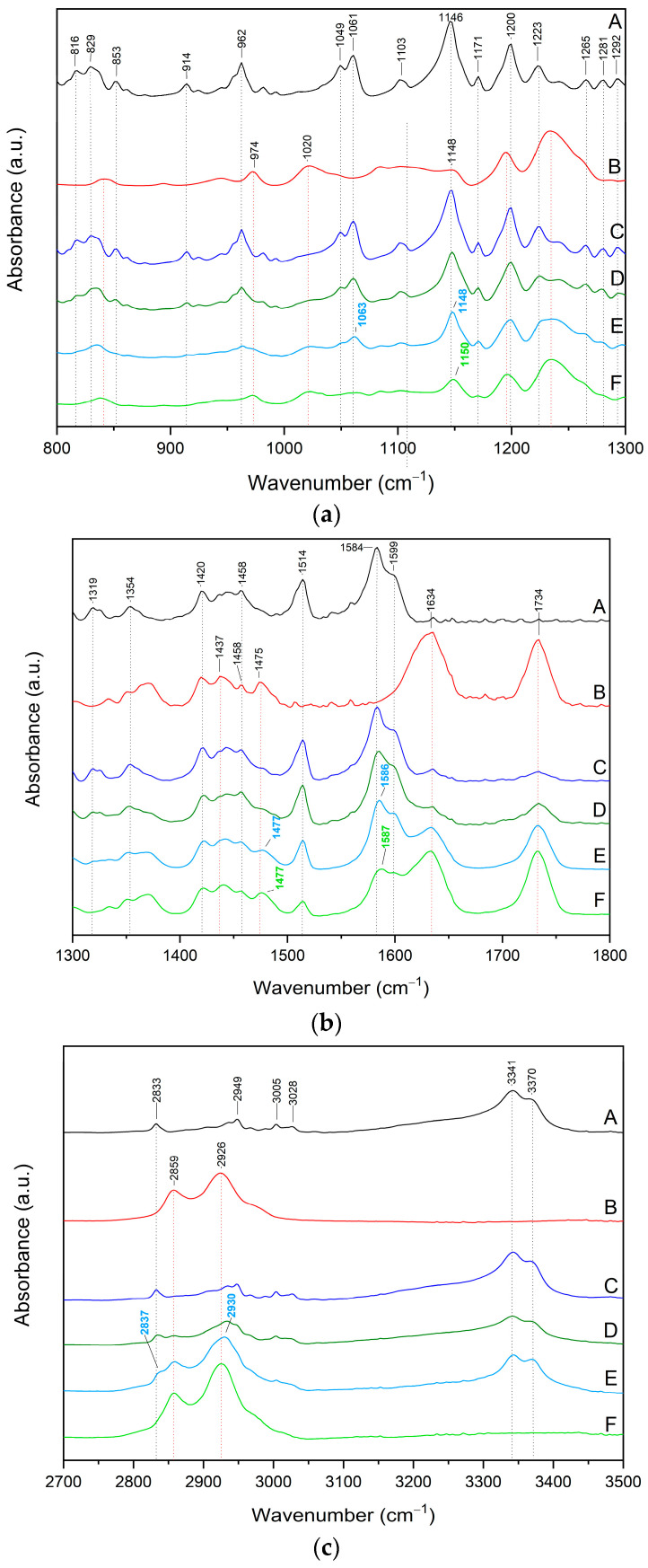
FT-IR-ATR analysis of pterostilbene (black line, A), Soluplus^®^ (red line, B), pterostilbene-Soluplus^®^ physical mixture 1:2 *w*/*w* (dark blue line, C), pterostilbene-Soluplus^®^ physical mixture 1:5 *w*/*w* (dark green line, D), pterostilbene-Soluplus^®^ system 1:2 *w*/*w* (blue line, E), and pterostilbene-Soluplus^®^ system 1:5 *w*/*w* (green line, F). (**a**) Range 800–1300 cm^−1^; (**b**) range 1300–1800 cm^−1^; (**c**) range 2700–3500 cm^−1^.

**Figure 8 pharmaceutics-15-01231-f008:**
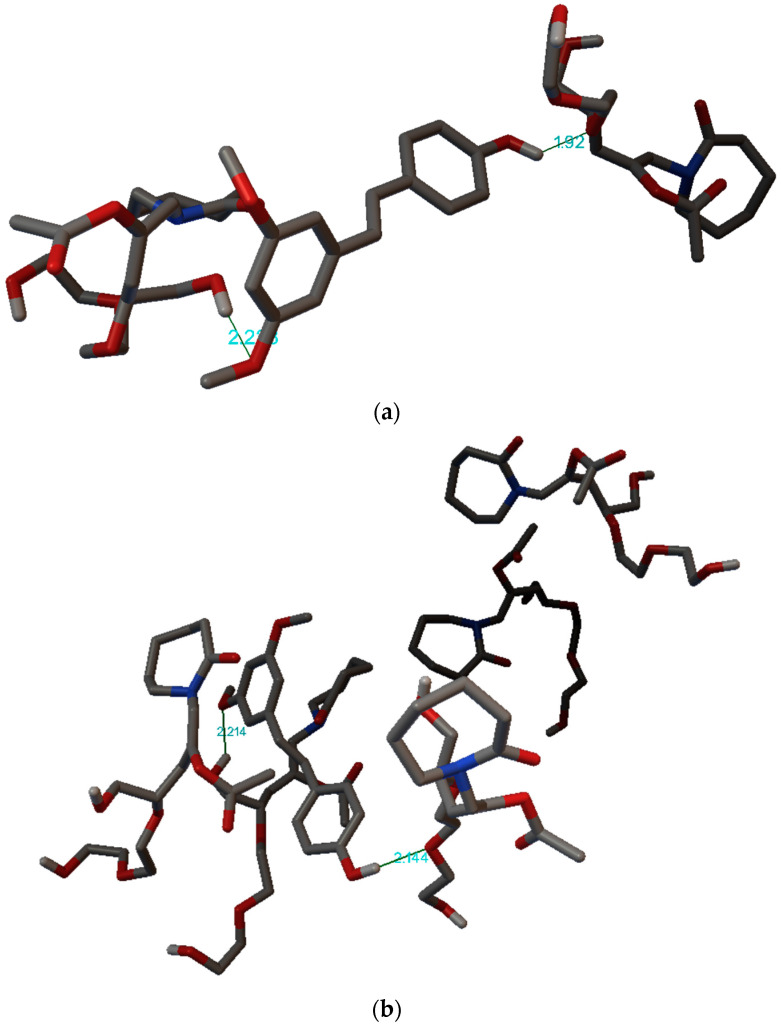
Three-dimensional view of pterostilbene-Soluplus^®^ interactions (hydrogen bonds) of the best poses generated in the Autodock Tool: (**a**) one structure of PTR and two monomer structures of SOL and (**b**) one structure of PTR and five monomer structures of SOL.

**Table 1 pharmaceutics-15-01231-t001:** Summary of the most important parameters of thermal analysis of PTR-SOL systems, and the experimental and theoretical T_g_ values of PTR-SOL systems.

	Mass (mg)	ΔC_p_ $\left(J\cdot {\left(g\cdot {}^{\circ}C\right)}^{-1}\right)$	T_g,exp_ (°C)	T_g,G-T_ (°C)	T_g,C-K_ (°C)	Deviation
PTR	7.2	0.461	12.9			
SOL	9.45	0.247	79.2			
PTR-SOL 1:2 *w*/*w*	7.97	0.229	59.8	30.2	50.7	+
PTR-SOL 1:5 *w*/*w*	6.34	0.188	65.3	43.7	63.6	+

ΔC_p_—heat capacity, T_g,exp_—glass transition temperature (experimental), T_g,G-T_—glass transition temperature (calculated by Gordon-Taylor equation), T_g,C-K_—glass transition temperature (calculated by Couchman-Karasz equation), positive deviation (+).

## Data Availability

The data are contained within the article and Appendix A.

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
