# Peer review of "Amorphous Pterostilbene Delivery Systems Preparation—Innovative Approach to Preparation Optimization"

_pharmaceutics, 2023, doi:10.3390/pharmaceutics15041231_

Round 1
Reviewer 1 Report
The manuscript entitled "Amorphous pterostilbene delivery systems preparation – innovative approach to preparation optimization", submitted by Natalia Rosiak et al, addresses the important issue of the bioavailability of natural active ingredients (PTR in this case) with low solubility in aqueous media. The authors propose "a novel amorphous solid dispersion (ASDs) of pterostilbene (PTR) with Soluplus® (SOL) to improve the solubility and antioxidant activity of PTR". There is an ample experimental effort complemented by a sizable computational one. However, the actual text does not serve these efforts.
The attached files contain several comments, corrections, suggestions for the improvement of various issues:
- elliptical, clumsy, confusing formulations; several typos;
- wrong/incomplete legends and/or errors in some figures
- wrong assertions in Conclusion; anyway, this section should be expanded and include emphasis on what the authors claim as originality features of their work.
With a thorough editing and appropriate corrections this manuscript would reach a publishable form.

Author Response
Dear Reviewer, thank you for your valuable suggestion and PDF file with comments. The authors have made all the suggested corrections. In the current version of the manuscript, the "Results" and "Discussion" sections have been combined into a single "Results and Discussion" section. At the suggestion of the editor, the authors have rewritten the section "Materials and methods". The English language was revised by a professional computer program and read by a mother-English tongue specialist.
In the attachment is PDF file with your comment and our answers.

Reviewer 2 Report
Rosiak et al. describe preparing and characterizing amorphous solid dispersions (ASD) of pterostilbene (PTR) with Soluplus (SOL) as a carrier polymer. The authors screened various w/w ratios between the components and determined that the 1:2 and 1:5 ratios exhibited complete amorphization. Using mathematical models, the experimental values ​​of Tg were correlated. Additionally, they carried out experiments to determine solubility and antioxidant activity, and the systems mentioned above showed considerable increases in these properties compared to pure PTR.
In general, I consider all research to be appropriate for publication in Pharmaceutics; however, before this happens, authors are requested to make the following changes:
1) In the case of FT-IR analysis, it is too much text. It would be much more beneficial for the reader if they concentrated all the vibration modes in a table. Where the corresponding vibration modes of the initial components and the two ASDs are indicated. And right there, make a description of the observed changes.
2) What is described in points 2 (results) and point 3 (discussion) seems to be very repetitive. Please summarize everything in one point. Much information was already described in point 2 and appeared again in point 3.
In this way, these major corrections are requested.Author Response
Please see the attachment.

Reviewer 3 Report
This manuscript presents a comprehensive study for amorphous solid dispersion design of PTR and SOL. Various analytical results including simulation were used. In addition, solubility and antioxidant study were finally demonstrated to show the beneficial effect of amorphous solid dispersion design. I recommend publishing this manuscript after minor revision. My suggestions are listed as bellows.
1. In Abstract, in addition to mention the solubility value, I recommend adding the times for solubility improvement for clearly comparing this formulation with neat API.
2. In section 2.2.3, please give an illustration for why a fast scanning rate up to 40 oC/min was used.
3. In section 2.6.2, the time for vortex mixing was 60 sec. I think the solubility value reported in this study should be “apparent solubility”, not the “equilibrium solubility”. Please give a precise definition of solubility used in this study.
4. In Figure 1, please mention the curves are 1st heating or 2nd heating. I think these curves should be 2nd heating since API did not show a crystalline property.
5. In Figure 4, please remove the “second heating” in the figure since the last two curves were data acquired from 1st heating.
6. Since a milling process was used for amorphous solid dispersion design, the solubility improvement may also attribute to the micronization. If available, please also give the SEM or PSD results for amorphous solid dispersion samples and neat API for comparison
Round 2
Reviewer 2 Report
This revised version has corrected previous requests made by a server. Therefore, this manuscript is suitable for publication in Pharmaceutics. However, consider that there are certain problems in the manuscript.
On page 6, in lines 170-175, the same sentence is repeated several times.
Again, on page 20, in lines 570-573, the same sentence is repeated several times.